*Experimental Results* (2020), 1, e41, 1–14

CAMBRIDGE
UNIVERSITY PRESS

**LIFE SCIENCE AND BIOMEDICINE**

NOVEL-RESULT

# A Computational Model for Estimating the Progression of COVID-19 Cases in the US West and East Coast Population Regions

Yao-Yu Yeo[1,*] ⓘ, Yao-Rui Yeo[2] and Wan-Jin Yeo[3,4]

[1]Department of Microbiology and Immunology, Cornell University, Ithaca, NY14850, USA, [2]Department of Mathematics, University of Pennsylvania, Philadelphia, PA19104, USA, [3]Department of Physics, University of Washington, Seattle, WA98195, USA; and [4]Institute for Learning and Brain Sciences, University of Washington, Seattle, WA98195, USA
*Corresponding author. E-mail: yy826@cornell.edu

(Received 08 July 2020; Revised 08 August 2020; Accepted 13 August 2020)

**Abstract**

The ongoing coronavirus disease 2019 (COVID-19) pandemic is of global concern and has recently emerged in the US. In this paper, we construct a stochastic variant of the SEIR model to estimate a quasi-worst-case scenario prediction of the COVID-19 outbreak in the US West and East Coast population regions by considering the different phases of response implemented by the US as well as transmission dynamics of COVID-19 in countries that were most affected. The model is then fitted to current data and implemented using Runge-Kutta methods. Our computation results predict that the number of new cases would peak around mid-April 2020 and begin to abate by July provided that appropriate COVID-19 measures are promptly implemented and followed, and that the number of cases of COVID-19 might be significantly mitigated by having greater numbers of functional testing kits available for screening. The model is also sensitive to assigned parameter values and reflects the importance of healthcare preparedness during pandemics.

**Keywords:** COVID-19; SARS-CoV-2; coronavirus; viral transmission; computational model

## Introduction

COVID-19 (previously 2019-nCoV) is caused by the new SARS-CoV-2 coronavirus and originated from Wuhan city, China (Zhu et al., 2020). Although the first case was reported in December 2019, 2020, COVID-19 has continued to spread around the world and over 150 countries have been affected to this day (WHO, 2020). COVID-19 was first reported in the US on January 20, 2020 (Holshue et al., 2020). While little action was taken initially, an exponential increase in the number of cases have spurred immediate actions to contain COVID-19. For instance, social distancing is being enforced via the closure of educational institutions, restrictions on travel, and suspension of events. Unfortunately, containment has been hindered by various factors such as the initial production of defective test kits and a current lack of test kits and medical supplies (Yeo & Ganem, 2020). As a result, it is currently unclear how well the US will cope with COVID-19 over the next few months, especially in view of the lack of adequate projections of the scale of COVID-19 infections and mortality across the country at the time of completing this study (March 19, 2020).

## Objective

In this paper, we attempt to construct a mathematical model that simulates the scale of COVID-19 outbreak in the West and East Coast population regions of the US.

## Methods

### Designing the Model

To model the COVID-19 outbreak in the US, we use a variant of the SEIR model (see (Dureau et al., 2013; Lin et al., 2020) for some examples). We mainly focus on the West and East Coast population regions of the United States. We assume that there is no travel between the different population zones because people living at the opposite ends of the coasts seldom intertravel and travels will eventually halt while the US transitions to lockdown. We also assume that the natural birth and death rates are equal because the total population does not differ significantly during a short timeframe in the absence of a catastrophe.

Figure 1 depicts the basis of our model, focusing on a particular coast, where arrows indicate the flow of the population at different stages.

In Fig. 1, $S$ represents the susceptible population, and $E$ represents the exposed population (i.e. individuals who have been infected but are not yet themselves infectious). The infected population $I$ is divided into two groups, $I_H$ and $I_C$, wherein the subscripts H and C stand for "hospital" and "community" respectively. $I_H$ represents those that are infected and isolated (such as those that have been pre-tested and found to carry the virus), and $I_C$ for those that are infected but not isolated (such as those with unreported cases or present mild symptoms that are overlooked). Thus, people in $I_H$ are unable to spread the virus whereas those in $I_C$ can spread the virus. While $I_H$ may potentially contribute to in-hospital transmissions to health professionals, the relative number of cases would be much lower and can be negligible because hospitals are equipped with personal protective equipment and proper sanitization, and health professionals represent a low number of the population. As testing kits in the US are currently in low supply, the current model projects significantly higher levels of $I_C$ than $I_H$ at any given point in time. Once infected, there are two possibilities: either recovery or death; these outcomes are represented by the populations $R_H$, $R_C$ and $D_H$, $D_C$ respectively. We also assume the recovered population will have acquired immunity to the virus and are no longer susceptible (Bao et al., 2020; Lu et al., 2020).

In summary, below are the key assumptions we used to design the model:

- no travel between coasts,
- equal birth and death rates,
- infected populations that arise from in-hospital transmissions are negligible,
- virus immunity after contracting COVID-19,
- $I_C = 1$ at the start (see implementation of model).

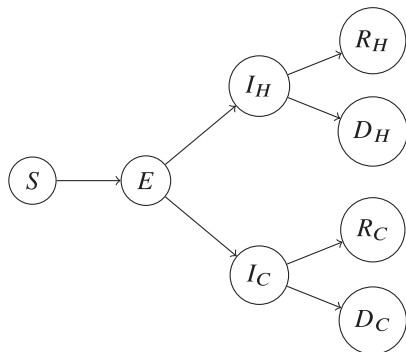

**Figure 1.** A diagram summarizing our modified SEIR model for each coast.

The actual mathematical model underlying Fig 1 is as follows.

$$\frac{dS}{dt} = -\beta\frac{I_C}{N}S \qquad \frac{dI_H}{dt} = \alpha\rho E - \gamma I_H \qquad \frac{dR_H}{dt} = \gamma(1-c)I_H \qquad \frac{dD_H}{dt} = \gamma c I_H$$

$$\frac{dE}{dt} = \beta\frac{I_C}{N}S - \alpha E \qquad \frac{dI_C}{dt} = \alpha(1-\rho)E - \gamma I_C \qquad \frac{dR_C}{dt} = \gamma(1-c)I_C \qquad \frac{dD_C}{dt} = \gamma c I_C$$

In the differential equations above, $N = S + E + I_H + I_C + R_H + R_C$ represents the *effective total population* in the US, i.e. the population that matters for the purposes of virus transmission. The various parameters in the differential equations and their chosen values (based on (Lauer et al., 2020; Lazzerini et al., 2020; Lin et al., 2020; Linton et al., 2020; Liu et al., 2020)) are also summarized in the table below. In particular, in our implementation we account for the variations of incubation period $\alpha^{-1}$ and infectious period $\gamma^{-1}$ by sampling 100 values from an Erlang distribution of shape 2 with minimum restricted to be 1 day (i.e. minimum possible incubation period is 1 day). The West Coast also takes the lower of the $R$ values to account for the lower average population density.

| Parameter | Definition | Value (and distribution if applicable) |
|---|---|---|
| $\alpha^{-1}$ | Incubation period | Erlang with shape 2 and mean 5.2 days |
| $\gamma^{-1}$ | Infectious period | Erlang with shape 2 and mean 3 days |
| $R^{(1)}$ | Reproduction rate in Phase 1 | 2.7 (West Coast); 2.9 (East Coast) |
| $R^{(2)}$ | Reproduction rate in Phase 2 | 2.3 (West Coast); 2.5 (East Coast) |
| $R^{(3)}$ | Reproduction rate in Phase 3 | 1 |
| $\beta$ | Contact rate per unit time | $R\gamma$ |
| $\rho$ | Proportion of exposed that are pre-tested | 0.1 |
| $c$ | Case fatality rate | 0.05 |

Notice that the reproduction rate $R$ is not present in the differential equations. However, $R$ is important even if the contact rate is hard to directly estimate, and we will use $R$ and $\gamma$ to estimate $\beta$ using the relation $\beta = R\gamma$. The reproduction rate is also not static over time; as hinted above in the table, we will use different values over the following three phases.

- Phase 1: There is no action done on the epidemic.
- Phase 2: Some action is done to slow down the epidemic while preparing for the next phase.
- Phase 3: Coast shutdown; schools moved online, events canceled, and most public areas closed.

To prepare for a quasi-worst-case-scenario, we assume $R \geq 1$, a necessary condition for any disease outbreak (Keeling et al., 2008). This is because $R = 1$ represents a neutral reproduction rate, whereas $R < 1$ and $R > 1$ represents a decline and increase of spread, respectively.

### *Implementing the Model*

We simulated our model in MATLAB to prepare for a quasi-worst-case-scenario. We employed a fourth-order Runge-Kutta method (Hubbard & West, 1991) to numerically solve the above specified ordinary differential equations, with the range $0 \leq t \leq 250$ and stepsize $h = 1/3$ (both measured in days). As for the initial conditions for each coast, we assume that a single infected person is introduced, and that there were no pre-existing immune responses that may help defend against the virus due to SARS-CoV-2 being sufficiently divergent from other CoVs (Bao et al., 2020; Lu et al., 2020). Hence, the entire population is initially susceptible due to the virus.

## Results

Using elements outlined in the previous sections to model the situation for the West and East Coast population regions, as well as the entire US, we ran 50 simulations using the model described above. We then compared our simulation results to current data between January 20, 2020 and March 19, 2020 (Kaggle, 2020), assuming a delay in reporting time of about 8 to 10 days and using population estimates derived from (US Census Bureau, 2020).

### The West Coast

The first case of COVID-19 in the West Coast was reported on January 20, 2020 in Seattle, WA. We used a population estimate of 53 million. If January 20, 2020 is designated Day 0, we assume Phase 2 started at around Day 45, and Phase 3 started at around Day 60. The simulation results suggest that, under quasi-worst-case-scenario, the data for the number of people infected by COVID-19 is far more than reported: about 80% of $I_C$ (those infected but not isolated) are not accounted for. This is probable as many symptoms of COVID-19 may be passed off as just cases of mild flu. In the quasi-worst-case-scenario, we predict the number of reported infections at its peak to be about 25,000 and the actual number of infections at about 90,000 (Fig. 2A).

### The East Coast

The first case of COVID-19 in the East Coast was reported on February 1, 2020 in Boston, MA. However, since no other new cases were reported for about two weeks, we assume a starting time of one month later than for the West Coast. In the graphs below, we will still plot them starting from the first reported case in the US (i.e. January 20, 2020). For the East Coast, we used a population estimate of 152 million. Based on government actions, we also assume that the starting dates for Phases 2 and 3 are about a week behind that of the West Coast. The simulation results are similar to that of the West Coast; for a quasi-worst-case scenario, the actual number of infections is predicted to be about 400,000 for reported, and about 1,450,000 for the actual number of infections (Fig. 2B).

### The United States: Hypothetical 25% Infected Worse-Case Scenario

As COVID-19 has escalated into a pandemic, we decided to compare to the previous 2009 Influenza (H1N1) pandemic. Although H1N1 and SARS-CoV-2 are distinct viruses with different virology and epidemiology, 2009 H1N1 pandemic influenza is the only other pandemic to have emerged in modern society and can help potentially help project the full extent of the spread of COVID-19. Considering that around 24% of people were infected during the 2009 H1N1 pandemic (Kerkhove et al., 2013), as well as the non-static nature of our parameters, we ran another simulation on the entire US population with approximately 22% to 26% of the population infected (based on randomness) to predict such a scenario. For this simulation experiment, the values we used for $R$ and the starting days of the phases was a rough weighted estimate of the two coasts. The values for the other parameters remained the same, as described in the previous section. We also assumed five infected individuals were introduced into the population. As Fig. 2 C illustrates, in this quasi-worst-case-scenario, about 3.8 million of the US population would die from COVID-19 assuming a 5% mortality rate, and the remaining 76.2 million would recover.

## Discussion

This paper presents a model for simulating the scale of COVID-19 pandemic in the United States. The simulations focused on the West and East Coast population regions (Figs. 2A, 2B) and also modeled a hypothetical worst-case scenario for the entire US (Fig. 2C). The model is different from other computational models because factors specific to the US are considered, such as the three phases of

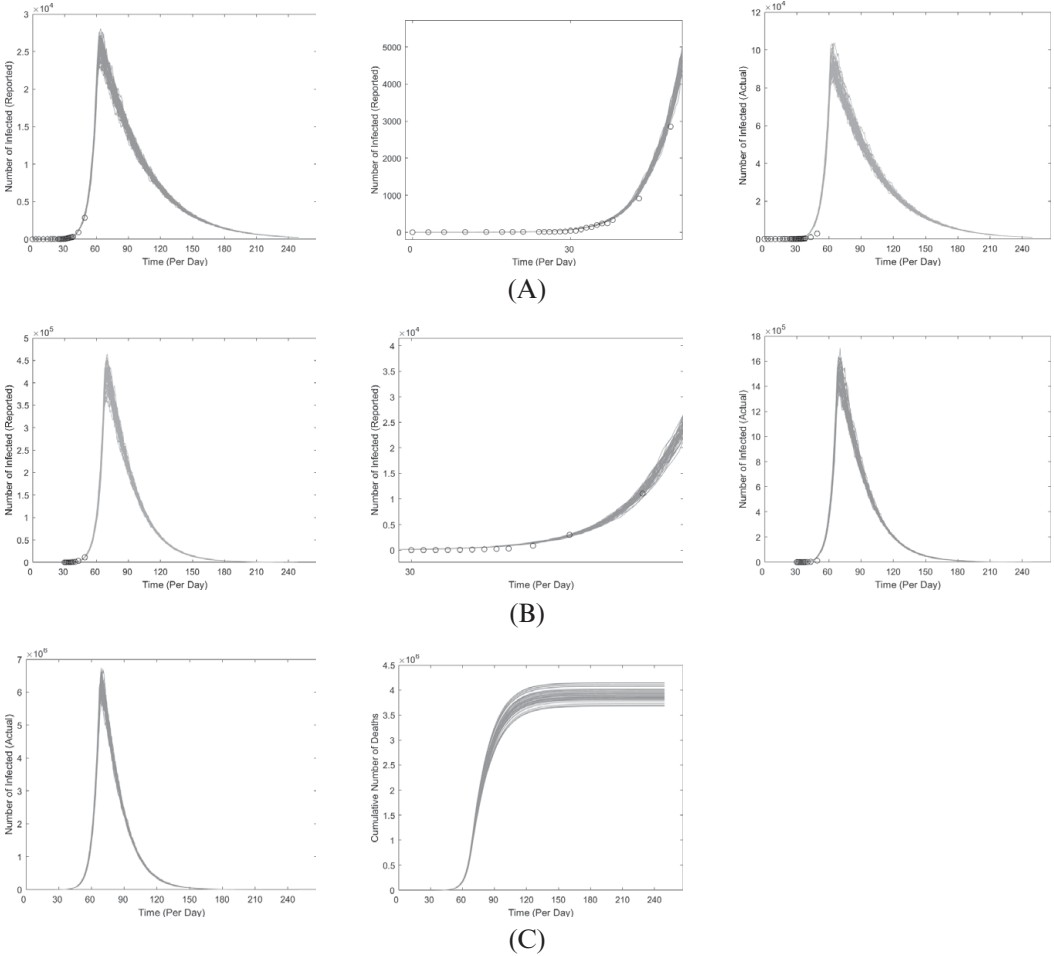

**Figure 2.** Simulations of COVID-19 in the West **(A)** and East **(B)** Coasts. On the x-axis, the starting date at the origin is January 12, 2020, and numbers represent the days that have since elapsed (each interval is approximately 1 month). Each line represents one simulation. **(A, B)**: *Left*: The estimated total number of reported infections $I_R$ over time using the formula $I_R = I_H + 0.2 I_C$; *Center*: A magnification that includes available data to date. The circles represent the actual data of reported cases (adjusted for delay); *Right*: The estimated number of total infections $I$ over time, using the formula $I = I_H + I_C$. The number of reported cases is much lower than the actual number of cases. **(C)**: A hypothetical scenario where 25% of the US will be infected; *Left*: the estimated number of infected people over time; *Right*: the cumulative number of deaths by considering a constant 5% mortality at each unit time.

government response, travel restrictions that restrict cross-boundary transmissions, limits to testing kits and healthcare accessibility resulting in semi-strict isolation, as well as different transmission dynamics during each response phase.

The model fits the reported data by assuming that about 80% of the non-isolated cases are not reported; however, the actual number of cases may be much higher (Figs. 2A, 2B). That assumption agrees with a prior study (Li et al., 2020) and provides another evidence that COVID-19 is not easy to contain. We arrived at an 8 to 10 days reporting delay in a method similar to (Li et al., 2020). While this is an important factor to consider in disease modeling, the actual reporting delay of COVID-19 is currently too early to tell, and reporting delays usually show very high variance for many reasons (e.g. nature of the disease, transparency of healthcare systems). It may be worthwhile to note that some common distinctive respiratory viral diseases were found to have a mean of 9 days in reporting delay elsewhere (Marinović et al., 2015), a value which aligns with our simulation estimates. The model also predicts that the peak of

the outbreak would occur by early April, and the outbreak would wind down by the start of July. However, the model does not account for delays in reporting; as such, one should expect the peak reported number of infections to occur around mid-April. The peak would also occur sooner on the West Coast, which was affected first. All peaks should occur around mid-April unless there are major changes to the reproduction rates (for which there is no current evidence), significant lapses during the handling of COVID-19 or maintaining social distancing. Otherwise, COVID-19 will only continue to progress for the worse in terms of scale and duration; for instance, the July cool-down might be delayed, or worse, the US could even face subsequent outbreaks reminiscent of historical outbreaks. It should be noted that the trends in the model's estimates share similarities with the epidemic curves in some nations during the SARS-CoV-1 outbreak (Wallinga et al., 2004).

One way to mitigate the spread of COVID-19 is to successfully isolate more infected people, i.e. to have a higher relative proportion of $I_H$ over $I_C$. That outcome should be achieved when defective testing kits are replaced and more testing kits are allocated, so that more exposed people can be tested and isolated accordingly. For instance, if 50% more people had been tested since the start of the outbreak, the model would predict 40% of the projected severity (Figs. 3A, 3B). As the US faced mishaps in the allocation of testing kits during the initial handling of COVID-19 (Yeo & Ganem, 2020), the original peak is likely inevitable. Testing more people from this point onwards (i.e. late March) would help accelerate the reduction; a simulation for is shown in Figs. 3C, 3D. Additional ways to mitigate the mortality of COVID-19 include improving accessibility to healthcare and boosting quantities of essential supplies such as masks, respirators, and ventilators. However, the sudden onset of COVID-19 and the overall unpreparedness of the United States make these data inaccessible (Yeo & Ganem, 2020). It should also be noted that while the currently global mortality of COVID-19 is 3.8%, it is noticeably different across nations (Kaggle, 2020; Lazzerini et al., 2020). As such, the actual case-fatality rate of COVID-19 in the US cannot be accurately determined currently until further progress of COVID-19. As a result, we chose a 5% mortality rate in our quasi-worst-case scenario prediction of COVID-19 in the US, which is the weighted average of top 5 mortality rates in countries with noticeable number of COVID-19 cases (>1500) as of March 18 (Lazzerini et al., 2020). It is also important to note that other factors (e.g. abiotic, immunological) can further modulate our parameters and affect our current projections. As a result, our study used data from prior studies based on other countries hit relatively hard by COVID-19 such as China and Italy, and enables us to construct a quasi-worst-case model of COVID-19 in the US.

Our model and projections can be further improved when transmission dynamics in the US begin to be documented and governmental response begin to be properly executed. As of today (August 8, 2020; manuscript revision), the trends observed initially bore resemblance to our simulations, albeit at a smaller scale as we are doing a quasi-worst case scenario. However, this has sharply deviated since the start of June where the US is currently experiencing a substantial surge of new cases, especially a spike of new cases in July that is twice the April peak (CDC, 2020). This can be attributed to many clear evidences that distancing measures have not been properly followed nor well-implemented. Some examples include several states actively pushing for re-opening despite warnings from health officials, social justice movements across the US, et cetera. In consideration of these unexpected events that have significantly perturbed the reproduction rate, we can introduce a "Phase 4" to account for this by increasing the reproduction rate in the East and West Coasts at Day 135 (see Supplementary Material).

We are mindful that computational simulations are, by their very nature, approximations. There are currently no predictive models that satisfactorily produce a picture of the spread or clinical impact of the disease, including ours, as too many factors can affect the spread of a disease. In our model, we consider the East and West Coasts of the US as strictly isolated regions with static total population. While this is not true in reality, minimal movements between the coasts is expected once distancing measures are implemented. Moreover, our predictions can vary noticeably with small variations in assumptions and assigned parameter values. To cite one example, minor changes in the distribution between $I_H$, $I_C$, and $R$ can dramatically affect the model's predictions (see Supplementary Material). Also, at the societal level, the length of time that the cohort of patients remain hospitalized is often unknown in the early stages of a pandemic and can greatly influence the use and deployment of medical supplies and personnel, thus

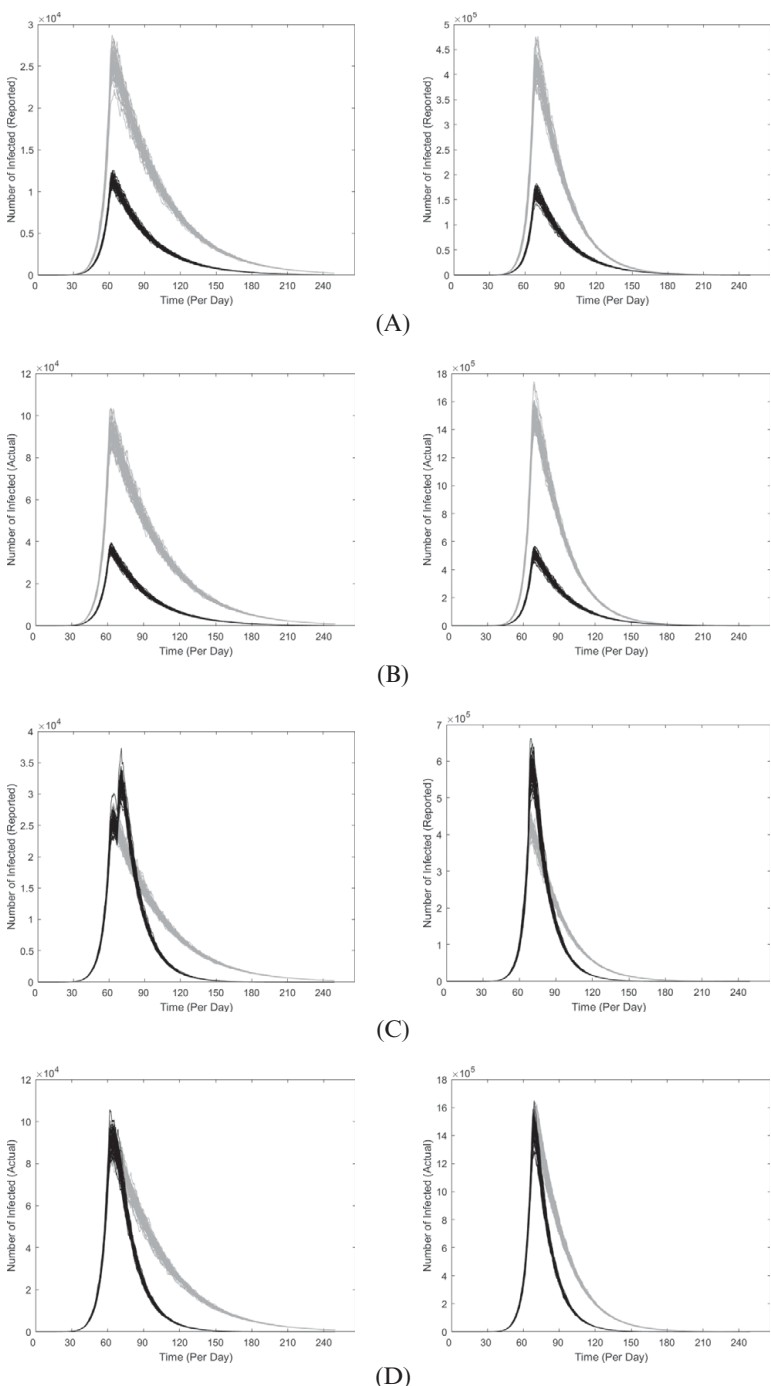

**Figure 3** Comparison of COVID-19 in the coastal US if more people were tested (solid) versus current projection (translucent). *Left*: West Coast; *Right*: East Coast. On the x-axis, the starting date at the origin is January 12, 2020, and numbers represent the days that have since elapsed (each interval is approximately 1 month). Each line represents one simulation. **(A)**: 50% more people tested since onset, considering only the infected that are reported. **(B)**: 50% more people tested since onset, considering the total number of infected. **(C)**: 3X the number of people tested starting late March, considering only the infected that are reported. Note that more reported cases would have been observed than originally projected because testing many more people shifts many potentially unreported IC to the IH cohort. Two peaks are observed for the West Coast (left), since the original peak is projected to have already passed, and implementing 3X more tests will result in another peak of reported infections. **(D)**: 3X the number of people tested starting late March, considering the total number of infected

altering the course of the infection and recovery rates. In addition, our test-and-isolate scenario is also quite optimistic as we assumed that hospitalized individuals can be fully isolated regardless of medical capacity despite a potential surge in COVID-19 patients. While some countries have been able to successfully contain COVID-19 by relying on efficient tests and isolation, it remains uncertain whether the US is able to similarly do so in light of constrains to testing, healthcare supplies, and capacities (Yeo & Ganem, 2020). Nevertheless, the few scenarios presented and discussed in this paper strongly suggest that even with current containment and mitigation efforts, COVID-19 outbreak will significantly impact the health of the US population in the East and West Coast.

## Conclusion

We constructed a modified SEIR computational model using current data that reflect the initial stages of COVID-19 in the US while considering different transmission dynamics across the anticipated phases of response to COVID-19 over the next months. Our current model projects that the number of new cases would peak around mid-April 2020, but our projections could potentially change over the course of COVID-19 in consideration of external factors (e.g. abiotic factors and governmental responses) that can significantly modulate transmission dynamics in the US, as well as the sensitivity of our model to adjustments in values assigned for each parameter. Future work will include adjusting the model accordingly to real-time COVID-19 situations (e.g. see Supplementary Material) and implementing the model to different disease outbreaks and other communities. We hope that this study provides an effective framework for modeling future disease outbreaks, help the mathematical modeling community to project COVID-19 and other past/present/future outbreaks, and make the nation more aware of the scope and nature of COVID-19.

**Acknowledgments.**  We thank Professor Bruce Ganem (Department of Chemistry and Chemical Biology) and Professor John Muckstadt (School of Operations Research and Industrial Engineering) of Cornell University for their encouragement, invaluable advice, and dedicated guidance throughout the study. We also thank Amandine Gamble (Department of Ecology and Evolutionary Biology) of UCLA for helpful comments on the manuscript.

**Author Contributions.**  YYY conceived the study and obtained the data. YRY and WJY designed and implemented the model and interpreted the results while YYY provided biological context. YYY, YRY and WJY wrote the manuscript. All authors contributed equally to the study.

**Funding Information.**  This research received no grants from any funding agency, commercial, or not-for-profit sectors.

**Data Availability Statement.**  The data used in the study are openly available at (Kaggle, 2020; US Census Bureau, 2020).

**Conflict of Interest.**  The authors declare that they have no competing interests.

**Supplementary Materials.**  To view supplementary material for this article, please visit http://dx.doi.org/10.1017/exp.2020.45.

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

# Peer Reviews

## Reviewing editor: Marc Henrion[1,2]

[1]Malawi-Liverpool-Wellcome Trust Clinical Research Programme, Statistical Support Unit, Queen Elizabeth Central Hospital, PO Box 30096, Blantyre, Malawi
[2]Liverpool School of Tropical Medicine, Clinical Sciences, Pembroke Place, Liverpool, United Kingdom of Great Britain and Northern Ireland, L3 5QA

This article has been accepted because it is deemed to be scientifically sound, has the correct controls, has appropriate methodology and is statistically valid, and has been sent for additional statistical evaluation and met required revisions.

doi:10.1017/exp.2020.45.pr1

## Review 1: A Computational Model for Estimating the Progression of COVID-19 Cases in the US West and East Coast Population Regions

**Reviewer:** Dr. Akira Endo 🆔

London School of Hygiene & Tropical Medicine, London, United Kingdom of Great Britain and Northern Ireland, WC1E 7HT

Date of review: 16 July 2020

**Conflict of interest statement.** Reviewer declares none

*Comments to the Author:* (Overall comment omitted due to word limit)
  - Evidence not available at time of writing: I understand, but neither the readers nor myself being able to read the cover letter, the author should either (i) update/validate their assumptions with the latest data to justify their conclusions, or (ii) clarify their analysis and conclusions were rather historical and might not be in line with the latest data. The paper may provide a case study (if technical issues below are addressed), but the conclusions are not justified unless the underlying assumptions are up-to-date.
  - Feasibility of test-and-trace: The predicted curves are 10 –100 times the SK and SG. I suggest focusing on isolation rather than hospitalization unless the bed occupancy is estimated using lengths-of-stay and medical capacity data.
  - Change reflecting R=1: I realized x-axis shows timesteps: please use unit of day.
  - 25% scenario: R0 determines the final size with the homogeneous-mixing assumption. Though R0 used for this scenario is not shown, I feel using the initial R throughout may be more appropriate for the quasi-worst-case-scenario.
  - Code: It is nice that the authors uploaded their code. I found two problems:
  1. 1/gamma and 1/alpha were drawn from Erlang distribution independently at each timestep; the resulting distribution of incubation/infectious periods are not following Erlang distribution.
  2. betaa is calculated for each individual as R×gamma[l], which suggests that individuals change their infectiousness according to their duration of infectiousness so that the product becomes constant R, which is implausible in actual transmission dynamics.

**Score Card**

Presentation

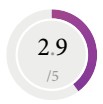

2.9
/5

| | |
|---|---|
| Is the article written in clear and proper English? (30%) | 4/5 |
| Is the data presented in the most useful manner? (40%) | 2/5 |
| Does the paper cite relevant and related articles appropriately? (30%) | 3/5 |

Context

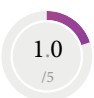

2.0
/5

| | |
|---|---|
| Does the title suitably represent the article? (25%) | 2/5 |
| Does the abstract correctly embody the content of the article? (25%) | 2/5 |
| Does the introduction give appropriate context? (25%) | 3/5 |
| Is the objective of the experiment clearly defined? (25%) | 1/5 |

Analysis

1.0
/5

| | |
|---|---|
| Does the discussion adequately interpret the results presented? (40%) | 1/5 |
| Is the conclusion consistent with the results and discussion? (40%) | 1/5 |
| Are the limitations of the experiment as well as the contributions of the experiment clearly outlined? (20%) | 1/5 |

doi:10.1017/exp.2020.45.pr2

# Review 2: A Computational Model for Estimating the Progression of COVID-19 Cases in the US West and East Coast Population Regions

**Reviewer:** Dr. James Maas ⓘD

Liverpool School of Tropical Medicine, Liverpool, United Kingdom of Great Britain and Northern Ireland, L3 5QA

Date of review: 20 July 2020

**Conflict of interest statement.** Reviewer declares none.

*Comments to the Author:* This manuscript describes a novel and potentially useful piece of work in modelling Covid-19 prevalence prediction. However its utility is somewhat limited, due in part to the rather terse response to previous reviewers comments. All models are comprised of assumptions, therefore to improve clarity, and reduce required speculation by the reader, please explicitly state as many assumptions, explicit and implicit, as possible. Input parameter estimate values present extremely valuable opportunities to perform "sensitivity analysis". A model found to be highly sensitive to specific parameters, suggests that your confidence in model predictions will be highly and directly correlated with the precision and confidence of these input parameters. You have hinted at some sensitivity analysis around the "25% infected worst-case scenario" however you have neither explained this precisely or provided the results of this analysis and thus have missed a very valuable opportunity to enlighten the reader. What results do you get with infected worst case scenario set at 5% increments from 15% to 35%? Could you do response analysis to number of days delay in reporting time, from 5 to 12 perhaps? The information highlighted in the figures could be presented more concisely in table form. How much did the time to peak, and number at peak vary across the 50 individual simulations? Your paper would benefit from considering these analyses because your model will produce them quickly.

---

## Score Card

### Presentation

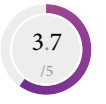

| | |
|---|---|
| Is the article written in clear and proper English? (30%) | 4/5 |
| Is the data presented in the most useful manner? (40%) | 4/5 |
| Does the paper cite relevant and related articles appropriately? (30%) | 3/5 |

3.7 /5

### Context

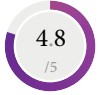

| | |
|---|---|
| Does the title suitably represent the article? (25%) | 5/5 |
| Does the abstract correctly embody the content of the article? (25%) | 5/5 |
| Does the introduction give appropriate context? (25%) | 4/5 |
| Is the objective of the experiment clearly defined? (25%) | 5/5 |

4.8 /5

## Analysis

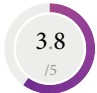

3.8
/5

| | |
|---|---|
| Does the discussion adequately interpret the results presented? (40%) | 4/5 |
| Is the conclusion consistent with the results and discussion? (40%) | 4/5 |
| Are the limitations of the experiment as well as the contributions of the experiment clearly outlined? (20%) | 3/5 |

doi:10.1017/exp.2020.45.pr3

# Review 3: A Computational Model for Estimating the Progression of COVID-19 Cases in the US West and East Coast Population Regions

**Reviewer:** Ivan Allaman 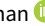

Universidade Estadual de Santa Cruz, Exact and Technological Sciences

Date of review: 04 August 2020

**Conflict of interest statement.** Reviewer declares none.

*Comments to the Author:* They are in an attached document.

---

## Score Card

### Presentation

**4.3** /5

| | |
|---|---|
| Is the article written in clear and proper English? (30%) | 5/5 |
| Is the data presented in the most useful manner? (40%) | 4/5 |
| Does the paper cite relevant and related articles appropriately? (30%) | 4/5 |

### Context

**4.5** /5

| | |
|---|---|
| Does the title suitably represent the article? (25%) | 5/5 |
| Does the abstract correctly embody the content of the article? (25%) | 5/5 |
| Does the introduction give appropriate context? (25%) | 3/5 |
| Is the objective of the experiment clearly defined? (25%) | 5/5 |

### Analysis

**4.0** /5

| | |
|---|---|
| Does the discussion adequately interpret the results presented? (40%) | 4/5 |
| Is the conclusion consistent with the results and discussion? (40%) | 4/5 |
| Are the limitations of the experiment as well as the contributions of the experiment clearly outlined? (20%) | 4/5 |