## [Reviewer Report · Review 1: A Computational Model for Estimating the Progression of COVID-19 Cases in the US West and East Coast Population Regions]

*Comments to the Author:* (Overall comment omitted due to word limit)

- Evidence not available at time of writing: I understand, but neither the readers nor myself being able to read the cover letter, the author should either (i) update/validate their assumptions with the latest data to justify their conclusions, or (ii) clarify their analysis and conclusions were rather historical and might not be in line with the latest data. The paper may provide a case study (if technical issues below are addressed), but the conclusions are not justified unless the underlying assumptions are up-to-date.

- Feasibility of test-and-trace: The predicted curves are 10 –100 times the SK and SG. I suggest focusing on isolation rather than hospitalization unless the bed occupancy is estimated using lengths-of-stay and medical capacity data.

- Change reflecting R=1: I realized x-axis shows timesteps: please use unit of day.

- 25% scenario: R0 determines the final size with the homogeneous-mixing assumption. Though R0 used for this scenario is not shown, I feel using the initial R throughout may be more appropriate for the quasi-worst-case-scenario.

- Code: It is nice that the authors uploaded their code. I found two problems:

1. 1/gamma and 1/alpha were drawn from Erlang distribution independently at each timestep; the resulting distribution of incubation/infectious periods are not following Erlang distribution.

2. betaa is calculated for each individual as R×gamma[l], which suggests that individuals change their infectiousness according to their duration of infectiousness so that the product becomes constant R, which is implausible in actual transmission dynamics.

## Score Card

### Presentation

2.9/5

Is the article written in clear and proper English?
30%
4/5

Is the data presented in the most useful manner?
40%
2/5

Does the paper cite relevant and related articles appropriately?
30%
3/5

### Context

2.0/5

Does the title suitably represent the article?
25%
2/5

Does the abstract correctly embody the content of the article?
25%
2/5

Does the introduction give appropriate context?
25%
3/5

Is the objective of the experiment clearly defined?
25%
1/5

### Analysis

1.0/5

Does the discussion adequately interpret the results presented?
40%
1/5

Is the conclusion consistent with the results and discussion?
40%
1/5

Are the limitations of the experiment as well as the contributions of the experiment clearly outlined?
20%
1/5

---

## [Reviewer Report · Review 2: A Computational Model for Estimating the Progression of COVID-19 Cases in the US West and East Coast Population Regions]

*Comments to the Author:* This manuscript describes a novel and potentially useful piece ofwork in modelling Covid-19 prevalence prediction. However its utilityis somewhat limited, due in part to the rather terse response toprevious reviewers comments. All models are comprised of assumptions, therefore to improve clarity, and reduce required speculation by thereader, please explicitly state as many assumptions, explicit andimplicit, as possible. Input parameter estimate values presentextremely valuable opportunities to perform "sensitivity analysis". Amodel found to be highly sensitive to specific parameters, suggeststhat your confidence in model predictions will be highly and directlycorrelated with the precision and confidence of these inputparameters. You have hinted at some sensitivity analysis around the"25% infected worst-case scenario" however you have neither explainedthis precisely or provided the results of this analysis and thus havemissed a very valuable opportunity to enlighten the reader. Whatresults do you get with infected worst case scenario set at 5%increments from 15% to 35%? Could you do response analysis to numberof days delay in reporting time, from 5 to 12 perhaps? The informationhighlighted in the figures could be presented more concisely in tableform. How much did the time to peak, and number at peak vary acrossthe 50 individual simulations? Your paper would benefit fromconsidering these analyses because your model will produce themquickly.

## Score Card

### Presentation

3.7/5

Is the article written in clear and proper English?
30%
4/5

Is the data presented in the most useful manner?
40%
4/5

Does the paper cite relevant and related articles appropriately?
30%
3/5

### Context

4.8/5

Does the title suitably represent the article?
25%
5/5

Does the abstract correctly embody the content of the article?
25%
5/5

Does the introduction give appropriate context?
25%
4/5

Is the objective of the experiment clearly defined?
25%
5/5

### Analysis

3.8/5

Does the discussion adequately interpret the results presented?
40%
4/5

Is the conclusion consistent with the results and discussion?
40%
4/5

Are the limitations of the experiment as well as the contributions of the experiment clearly outlined?
20%
3/5

---

## [Reviewer Report · Review 3: A Computational Model for Estimating the Progression of COVID-19 Cases in the US West and East Coast Population Regions]

*Comments to the Author:* They are in an attached document.

## Score Card

### Presentation

4.3/5

Is the article written in clear and proper English?
30%
5/5

Is the data presented in the most useful manner?
40%
4/5

Does the paper cite relevant and related articles appropriately?
30%
4/5

### Context

4.5/5

Does the title suitably represent the article?
25%
5/5

Does the abstract correctly embody the content of the article?
25%
5/5

Does the introduction give appropriate context?
25%
3/5

Is the objective of the experiment clearly defined?
25%
5/5

### Analysis

4.0/5

Does the discussion adequately interpret the results presented?
40%
4/5

Is the conclusion consistent with the results and discussion?
40%
4/5

Are the limitations of the experiment as well as the contributions of the experiment clearly outlined?
20%
4/5